# Effectiveness on Quality of Life and Life Satisfaction for Older Adults: A Systematic Review and Meta-Analysis of Life Review and Reminiscence Therapy across Settings

**DOI:** 10.3390/bs13100830

**Published:** 2023-10-11

**Authors:** Qing Zhong, Cheng Chen, Shulin Chen

**Affiliations:** Department of Psychology and Behavioral Sciences, Zhejiang University, No. 866 Yuhangtang Road, Hangzhou 310058, China

**Keywords:** ageing, active ageing, reminiscence therapy, life review, quality of life, life satisfaction

## Abstract

Background: With the growing trend of ageing, there is an urgent need for effective interventions that enhance positive psychological functions among older adults. Objective: (1) To evaluate the effectiveness of life review and reminiscence therapy in enhancing the quality of life and life satisfaction among older adults. (2) To discover efficacious variables during interventions, such as form of intervention and number of sessions. Methods: Relevant randomized controlled trials in both English and Chinese languages were searched across eight databases. The meta-analysis was conducted by a random effects model using STATA 17. The registration number of this review is CRD42023424085. Results: Thirty-two studies with 2353 participants were included. Experimental groups of older adults significantly improved their quality of life (SMD 1.07; 95% CI 0.48 to 1.66; *p* < 0.001) and life satisfaction (SMD 1.12; 95% CI 0.63 to 1.60; *p* < 0.001). Subgroup analyses revealed that individual sessions of life review and reminiscence therapy had a more significant impact on improving quality of life and six to eight intervention sessions could enhance life satisfaction more effectively. Conclusions: Life review and reminiscence therapy hold promise for application in medical and nursing care for older adults, suggesting the potential benefits of implementing intervention designs with effective settings for positive psychological functions.

## 1. Introduction

With life expectancy increasing, the proportion of individuals aged 60 and over is growing at a faster pace compared to any other age group. By the year 2030, it is forecast that approximately one-sixth of the global population will be individuals aged 60 years or older; by 2050, the global population of individuals aged 60 years and older will have doubled, reaching a count of 2.1 billion [1]. Therefore, ageing is becoming an inevitable global demand that requires more effective solutions across various living settings. In 2002, the World Health Organization adopted “active ageing” as a response to the challenges posed by population ageing [2]. As a multidimensional concept, active ageing is evaluated using a combination of objective and subjective indicators [3]. This process aims to optimize opportunities for health, participation, and security to enhance the quality of life as people age [2]. Consequently, as a state of being, quality of life would be a fitting objective indicator for assessing active ageing. Moreover, life satisfaction, as a fundamental dimension of subjective well-being, is evaluated from both the experience of pleasure and contentment with one’s own life. Thus, it would be an appropriate subjective indicator to assess the process of active ageing [4].

As individuals progress in age, the connection between psychological states and health grows increasingly significant [5]. In order to enhance the positive psychological functions of older adults, a range of psychological interventions has been extensively implemented across various living environments. After Robert Butler [6] initially introduced the concepts of life review and reminiscence as tools to help older adults comprehend their current circumstances by establishing connections with past experiences, life review and reminiscence therapy were developed and are currently widely employed as interventions in the care of older adults across various illnesses and settings, including nursing homes and hospitals [7]. Because of their shared theoretical foundations, the terms life review and reminiscence therapy are frequently employed interchangeably in interventions. Consequently, the synthesis of evidence from life review and reminiscence therapy among older adults assumes particular significance [8].

Up until now, a range of studies have supported the effectiveness of life review and reminiscence therapy in treating various psychological problems as cost-effective, non-pharmacological interventions with minimal side effects [9,10]. In a Bayesian network meta-analysis, the findings indicate that life review emerges as the best-ranked intervention for relieving anxiety and distress in an inpatient palliative care setting compared to three other short-term psychological interventions (cognitive behavioral therapy; mindfulness intervention; action control group) [11]. Moreover, another meta-analysis supports the idea that reminiscence therapy effectively improves cognitive function and alleviates depression in people with dementia (*p* < 0.001) [12].

However, the positive psychological outcomes associated with life review and reminiscence therapy lack consistent conclusions. A meta-analysis including 22 studies and 1972 participants revealed no statistically significant effect of reminiscence therapy on quality of life (*p* = 0.16) [13]. Moreover, another meta-analysis indicated no significant difference in the observed improvement in life satisfaction between the life review group and the control group (*p* = 0.14) [14]. Recently, two systematic reviews have concluded that reminiscence therapy is effective in improving quality of life and life satisfaction among the elderly in the community or older adults with intact cognition and mild cognitive impairment [15,16]. Nevertheless, the living environments and health statuses of older adults vary significantly. Therefore, there is an urgent need for a comprehensive quantitative assessment of intervention effectiveness, considering the varied living conditions of older adults and their diverse health statuses.

In summary, a lack of consistent findings regarding the efficacy of life review and reminiscence therapy in different settings in positive psychological outcomes exists, potentially impeding the optimal allocation of care resources. Consequently, to promote the process of active ageing, the objectives of this meta-analysis are to evaluate the effectiveness of life review and reminiscence therapy on the quality of life and life satisfaction of older adults, while also discovering the potential efficacious variables related to the settings. 

## 2. Methods

This review was conducted following the Preferred Reporting Items for Systematic Review and Meta-Analyses (PRISMA) guidelines [17]. The review protocol was registered with PROSPERO (registration number CRD42023424085) [18].

### 2.1. Search Strategy

We conducted a systematic literature search on 17th March 2023 across eight databases, including Web of Science, EBSCO, Embase, The Cochrane Library, ProQuest Dissertations & Thesis Global, WANFANG, VIP, and CNKI (the latter three were used for Chinese studies), which were established in 1997, 1944, 1988, 1993, 1743, 1993, 2000, and 1999, respectively. There were no restrictions regarding the publication date. Only studies in English or Chinese were reviewed in this meta-analysis. Both published articles and peer-reviewed dissertations were included. The following search terms were used: (reminiscence* OR “life review*” OR nostalgia*) AND (intervention* OR treatment* OR therapy* OR program*) AND (elderly* OR “old people” OR “older adult*” OR “the aged” OR “late life”). The search terms are created by Boolean logic retrieval. Additionally, the reference lists of eligible review articles were also considered.

### 2.2. Study Selection

The retrieved articles were imported into EndNote version 20 software to remove duplicates and subsequently imported into Rayyan (https://rayyan.ai (accessed on 17 March 2023)) for screening. Two authors (Q.Z. and C.C.) first screened the studies for initial inclusion based on titles and abstracts after excluding duplicates. The same two authors then independently identified studies meeting the inclusion criteria based on the full-length text. Any discrepancies between the two authors during the literature screen and full-text review were resolved through discussions with the third author (S.L.C.). The inclusion criteria were as follows: (1) older adults aged 55 years and above; (2) life review or reminiscence therapy as the intervention; (3) randomized controlled trials (RCTs) as the study design; (4) one or a combination of the two outcomes: quality of life and life satisfaction; (5) if the data were published repeatedly, studies reporting the data in more detail were included. The exclusion criteria were as follows: (1) non-research articles: conference articles and reviews; (2) unable to obtain the full text; (3) no sufficient information for computing effect sizes.

### 2.3. Data Extraction

After screening all studies, records of included studies were saved in Excel. A data charting form for recording relevant information was created and pilot-tested by the review team. Two reviewers (Q.Z. and C.C.) used the initial draft of the data charting form to code data and then revised the coding form based on a discussion with the review team. Two trained coders (Q.Z. and C.C.) independently completed the final coding form for each eligible paper. If there were differences in the coding of specific items, the third author (S.L.C.) would make the final decision. Studies were coded by methodological characteristics, including first author, year of publication, country, age, the proportion of females, sample size, health condition, intervention details (e.g., intervention types and intervention conductors), the frequency and duration of each intervention session, the length of the intervention given, the post-intervention follow-up period, and outcomes. If the article data contained SE, the formula (SE = SD/√n) was used to convert the data to SD. Authors were contacted via email if any data were missing from the study.

### 2.4. Quality Assessment

The risk of bias in the included articles was assessed by two reviewers (Q.Z. and C.C.) according to the criteria recommended by the Cochrane Handbook using the Cochrane risk-of-bias tool (RoB2) [19]. This tool assesses bias arising from (1) the randomization process (including random sequence generation and allocation sequence concealment); (2) intervention deviation (blinding of participants and personnel); (3) missing outcome data (risk of bias of the data and results calculation); (4) measurement of outcomes (risk of bias of the assessment methods); and (5) selection of results (risk of bias in selection of the reported result). Each of these domains is ranked for risk of bias as ‘low’, ‘some concern’, or ‘high’, with an overall assessment of study quality compiled. The results of the quality assessment were documented in Excel.

### 2.5. Data Synthesis and Analysis

Statistical analysis was conducted using STATA/MP software version 17.0. The adjusted mean difference (Hedges’ g), which could provide a more conservative estimation with small sample sizes, was used as the outcome statistic, and 95% confidence intervals were computed. The true difference between studies would affect the measurement of study variability, so analyses of impact sizes were undertaken with random effect models. The I^2^ statistic was utilized to evaluate the level of heterogeneity, with values of 25%, 50%, and 75% representing low, medium, and high levels of heterogeneity, respectively. Sensitivity analysis was performed to assess the influence of each individual study by systematically removing each study one at a time. To identify the factors contributing to heterogeneity (I^2^ ≥ 50%), subgroup analyses were conducted with Review Manager 5.4, incorporating three variables: form of interventions, total sessions of interventions, and types of comparison. Additionally, as a continuous variable, the year of publication was evaluated by meta-regression. To assess the presence of publication bias, the visual assessment of the funnel plot and Egger’s regression intercept were undertaken, if ten or more studies were included.

## 3. Results

### 3.1. Study Selection

A total of 4196 records were identified from eight databases. In addition, 33 relevant articles were identified from the reference lists of relevant reviews and selected articles. After removing the duplicates, the total number of records was 2401. Subsequently, 2278 records were excluded after the title and abstract screening, and the remaining 171 records were evaluated by the full texts. Finally, 139 studies were excluded based on the inclusion criteria and the remaining 32 studies were selected for inclusion in the final meta-analysis. The selection flow was presented in the PRISMA flowchart (Figure 1).

### 3.2. Study Characteristics

Among the 32 articles included in the meta-analysis, 37 RCTs involving 2353 older adults were conducted. The sample sizes ranged from 12 to 202 participants per study, and all of them comprised both male and female individuals. Specifically, females accounted for 62.79% of the total sample size. In this study, 37.44% (n = 881) of participants were diagnosed with different types of dementia and cognitive impairment; and 17.42% (n = 410) with psychological symptoms, including depression and anxiety. Furthermore, 13.34% (n = 314) of the participants were stroke patients; 5.48% (n = 129) with frailty; and 5.10% (n = 120) had chronic diseases.

Among the studies included in the analysis, fifteen were conducted in China, five in the United States, two in the United Kingdom, and one study each was conducted in Iran, Ireland, Japan, the Netherlands, Northern Cyprus, Portugal, Spain, Switzerland, and the Dominican Republic. Most of the included interventions (n = 21) were conducted in institutions such as hospitals, nursing homes, and day-care centers; only a few (n = 4) were conducted at home. Interventions were conducted by therapists (n = 11), researchers (n = 11), and nurses (n = 6). Life review and reminiscence therapy were implemented for varying durations, ranging from four weeks to thirteen weeks, and 30 min to 120 min for each session, with the majority consisting of eight weekly sessions. Most studies employed life review and reminiscence therapy by discussing specific themes, such as childhood experiences. Furthermore, meaningful items, such as photographs, songs, foods, or letters, were utilized as clues during the therapy sessions. Study characteristics are presented in detail in Table 1.

### 3.3. Risk of Bias

In terms of risk of bias (Figure 2), 80% of the included studies were rated as some concerns, and 10% were rated as low risk and high risk, respectively. As only RCTs that provided valid data were included in this meta-analysis, the risk of randomization bias (100%) and results reporting bias (90%) were both low. However, given that the two outcomes in this study were assessed through self-assessment scales, there were potential concerns and risks regarding deviations from intended interventions (65%) and the measurement of outcomes (80%).

### 3.4. Main Outcomes

#### 3.4.1. Quality of Life

Fifteen RCTs, involving 1280 participants, reported the effects of life review and reminiscence therapy on quality of life using the Quality of Life in Alzheimer’s Disease (Qol-AD) [27,37,39,42,45,46,47], the Control, Autonomy, Self-realisation and Pleasure Quality of Life questionnaire (CASP-19) [20], the Dementia Quality of Life (DQoL) [50], the Euroqol questionnaire (EQ-5D) [32], the Generic Quality of Life Inventory-74 (GQoLI-74) [25], the Karnofsky Performance Scale (KPS) [36], the 36-item Short-Form (SF-36) [38], the World Health Organization Quality of Life Questionnaire (WHOQOL_BREF) [44], and the World Health Organization Quality of Life (WHOQOL-100) [23]. The forest plot (Figure 3) showed a significant effect of life review and reminiscence therapy for improving quality of life among older adults (SMD 1.07; 95% CI 0.48 to 1.66; I^2^ = 95.50%; *p* < 0.001).

#### 3.4.2. Life Satisfaction

Twenty-two RCTs, involving 1098 participants, examined the effects of life review and reminiscence therapy on life satisfaction using the Life Satisfaction Index-A (LSI-A) [21,22,24,29,30,31,33,34,35,41,43,48,49,51], the Satisfaction With Life Scale (SWLS) [20,40], the Life Satisfaction in the Elderly Scale (LSES) [28], and the Revised Philadelphia Geriatric Center Morale Scale (RPGCMS) [26]. The results showed that significant standardized mean differences existed in life review and reminiscence therapy compared with controls for life satisfaction (SMD 1.12; 95% CI 0.63 to 1.60; I^2^ = 91.90%; *p* < 0.001) (Figure 4).

### 3.5. Sensitivity Analysis

Due to high heterogeneity, two sensitivity analyses were conducted to examine the potential outliers in the analyses of quality of life and life satisfaction. According to the results (Figure 5) for quality of life, Gan [27] and Li [36] were removed individually, and the effect size was significant, but heterogeneity was still high (SMD 1.26; 95% CI 0.74 to 1.78; I^2^ = 93.70%; *p* < 0.001; SMD 0.88; 95% CI 0.33 to 1.42; I^2^ = 94.50%; *p* < 0.001). The results for life satisfaction are similar (Figure 6); after removing Han et al. [30], the difference was significant, although heterogeneity remained high (SMD 0.85; 95% CI 0.48 to 1.22; I^2^ = 85.50%; *p* < 0.001).

### 3.6. Meta-Regression and Subgroup Analysis

Meta-regression and subgroup analysis were then conducted respectively for continuous variables and categorical variables to investigate the potential drivers of heterogeneity.

#### 3.6.1. Quality of Life

The meta-regression analysis revealed that the publication year of the studies was unable to explain the heterogeneity (*p* = 0.65). However, a significant difference (*p* = 0.006) was found in the subgroup analysis concerning the form of interventions (Table 2). Specifically, the subgroup analysis indicated that implementing life review and reminiscence therapy in a group setting did not yield a significant effect size on quality of life (SMD 0.18; 95% CI −0.15 to 0.52; I^2^ = 34.40%; *p* = 0.28). Furthermore, no significant differences were found in terms of the effect of the number of intervention sessions on quality of life among older adults (*p* = 0.71). Since only one study implemented no treatment as a control group, we were unable to assess the effects of different comparison groups for quality of life.

#### 3.6.2. Life Satisfaction

According to the meta-regression analysis, the publication year of the studies did not significantly contribute to the heterogeneity of the life satisfaction analysis (*p* = 0.28). In the subgroup analyses (Table 3), there were no significant differences in the form of interventions (*p* = 0.69) and the type of comparisons (*p* = 0.06) on life satisfaction among older adults. However, there was a significant difference in the number of intervention sessions on life satisfaction (*p* = 0.006). The findings indicated that there was no significant improvement in life satisfaction when the intervention was implemented in more than eight sessions (SMD 0.44; 95% CI 0.10 to 0.77; I^2^ = 37.50%; *p* = 0.14).

### 3.7. Publication Bias

A visual examination of the funnel plot for the included studies revealed an asymmetry in both quality of life (Figure 7) and life satisfaction (Figure 8). However, Egger’s regression intercept analysis indicated the absence of publication bias for quality of life (*p* = 0.13), while publication bias was detected for life satisfaction (*p* = 0.045).

## 4. Discussion

### 4.1. Main Findings

This meta-analysis revealed significant improvements in the quality of life and life satisfaction of older adults after receiving life review or reminiscence therapy, compared to the control groups. Consequently, implementing these two interventions in care and medical contexts holds promise for effectively promoting active ageing and yielding potential benefits for both individuals and societies. As for the long-term effect of life review and reminiscence therapy, five studies reported the results of quality of life four weeks (n = 2), six weeks (n = 1), and twelve weeks (n = 2) after the completion of the interventions. Acquiring a sufficient amount of follow-up data for research involving older adults poses a challenge. Consequently, due to the limited number of studies and the wide range of time periods employed, the studies included did not evaluate the long-term impact of life review and reminiscence therapy on quality of life. As for life satisfaction, 10 of 22 trials conducted a repeated evaluation at four weeks (n = 2), five weeks (n = 1), six weeks (n = 2), twelve weeks (n = 4), and forty-eight weeks (n = 1). The results showed that the effect of life review and reminiscence therapy on life satisfaction remained statistically significant even after 12 weeks following the completion of the intervention (SMD 1.59; 95% CI 0.39 to 2.80; I^2^ = 94.9%; *p* = 0.01). This result reminds us that incorporating reminiscence-based therapy alongside a comprehensive caring program could stimulate and sustain a positive mood among older adults, which could be beneficial to process active ageing.

However, a high level of heterogeneity was observed in the analysis of the results. Sensitivity analysis indicated the presence of outliers in the reporting of findings by certain studies. For instance, Gan [27] reported a significantly higher quality of life in the experimental group compared to the control group, contrary to what the data indicated. This reminded us to carefully assess the plausibility of data when selecting studies. Moreover, the variation in the results could be attributed to different interpretations of the outcome concept. Quality of life is a multifaceted variable, leading to different definitions and consequently varied measurement tools [52]. Because of the diversity of cultural backgrounds and social customs across various countries and regions, the applicability of a single scale to all situations is not feasible [53]. For instance, while the SF-36 is a widely used health-related survey [54] and is more suitable for elderly people in the community, the WHOQOL-100 places greater emphasis on individuals’ subjective perceptions of their life circumstances within the culture and values that shape their existence [55] and is mainly used for patients with Alzheimer’s disease and cancer. In conclusion, as previously elucidated, the differences in study design or measurement tools may contribute to the heterogeneity observed in the analysis of the results.

### 4.2. Implications for Future Studies

Subgroup analyses revealed that the impact on quality of life was influenced by the intervention setting. Specifically, when life review or reminiscence therapy was conducted in a group setting, the effect was not statistically significant. Additionally, another review also reported that individual sessions of reminiscence therapy are associated with benefits for cognition and mood among people with dementia [13]. However, Xu et al. [56] discovered that the effect size of individual reminiscence therapy was smaller compared to that of group sessions among older adults without obvious cognitive impairment. Researchers have suggested that a patient-centered intervention could be beneficial for people with dementia [7,57]. Therefore, in future studies, it is advisable to initiate the assessment of cognitive function, followed by tailoring the form of the intervention based on cognitive level. For example, individual forms of life review or reminiscence therapy may be efficacious for older adults whose cognitive abilities have experienced a decline.

The duration of the interventions had an impact on the significance of the effectiveness of life satisfaction. Findings from this study indicated that the effect was not significant when the intervention extended beyond eight sessions in total. Considering the potential fatigue experienced by older adults, a duration of six to eight weeks may be a reasonable range to set when designing interventions [58]. The characteristics of the intervention itself and the ethical requirements of informed consent have led to some risks of bias in this research question, both in terms of experimental intent and outcome measures. Thus, future research should employ more rigorous study designs, with an emphasis on blinded outcome evaluation and allocation concealment. Additionally, a limited number of variables were available for applications in subgroup analyses, as many studies lacked full details regarding the intervention design. As a result, it is helpful to discuss the mechanisms that underlie the effectiveness of the intervention if future studies report the comprehensive study design, such as the implementer and the detailed intervention plan for each session.

### 4.3. Strengths and Limitations

This meta-analysis assessed improving quality of life and life satisfaction by life review and reminiscence therapy in older adults, which shifted from the traditional perspective of alleviating negative emotions to improving positive ones. Previous research has shown that quality of life and life satisfaction serve as robust predictors of an individual’s health. Furthermore, fostering a positive attitude has been linked to enhanced physical and mental well-being [59]. Meanwhile, this study’s emphasis on enhancing these positive psychological functions is congruent with the objectives of active ageing, potentially empowering older individuals in their pursuit of holistic well-being [3,60].

This study has several limitations. Firstly, due to the methodological constraints inherent in a systematic review and meta-analysis, there are some unavoidable risks of bias. These biases are attributed to factors such as language restrictions, issues related to informed consent, or experimental intent, especially when working with older adults with cognitive impairments. Secondly, due to the high heterogeneity of this meta-analysis, more high-quality studies need to be included in the future to demonstrate the credibility of the results. Finally, the long-term effects of interventions and the working mechanisms were not sufficiently explored due to the limited information.

## Figures and Tables

**Figure 1 behavsci-13-00830-f001:**
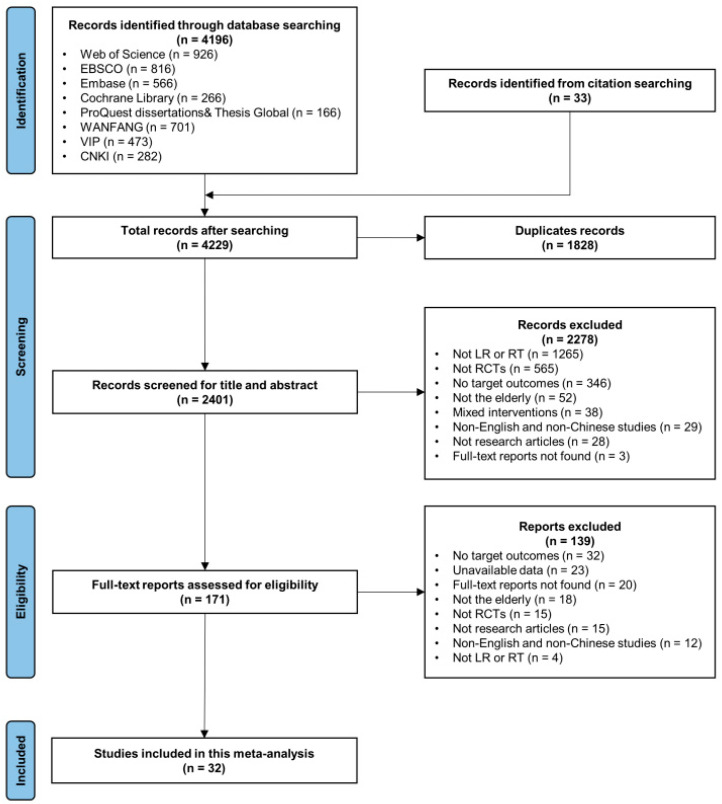
PRISMA flow chart.

**Figure 2 behavsci-13-00830-f002:**
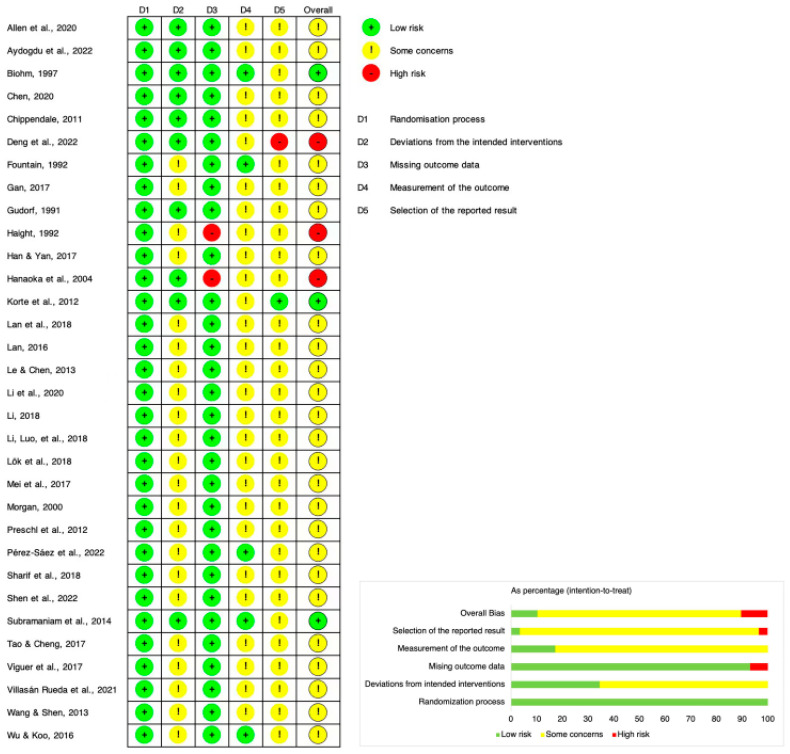
Risk of bias graph [20,21,22,23,24,25,26,27,28,29,30,31,32,33,34,35,36,37,38,39,40,41,42,43,44,45,46,47,48,49,50,51].

**Figure 3 behavsci-13-00830-f003:**
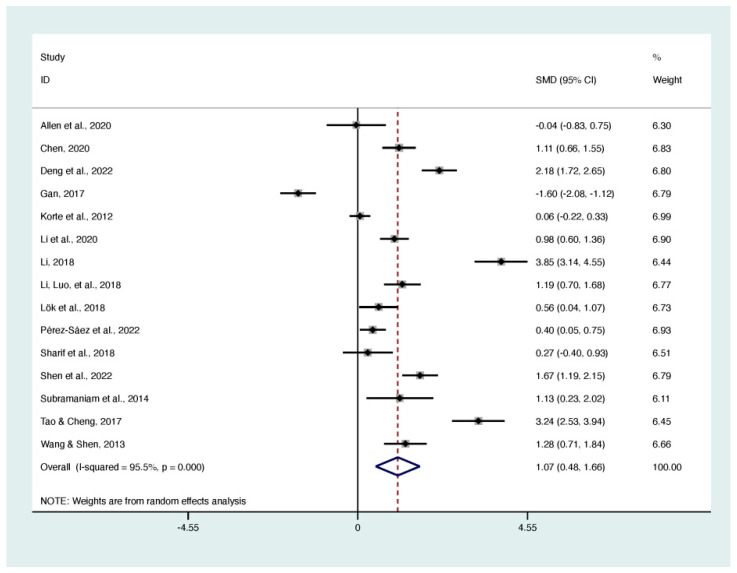
Forest plot of the effect on quality of life [20,23,25,27,32,36,37,38,39,42,44,45,46,47,50].

**Figure 4 behavsci-13-00830-f004:**
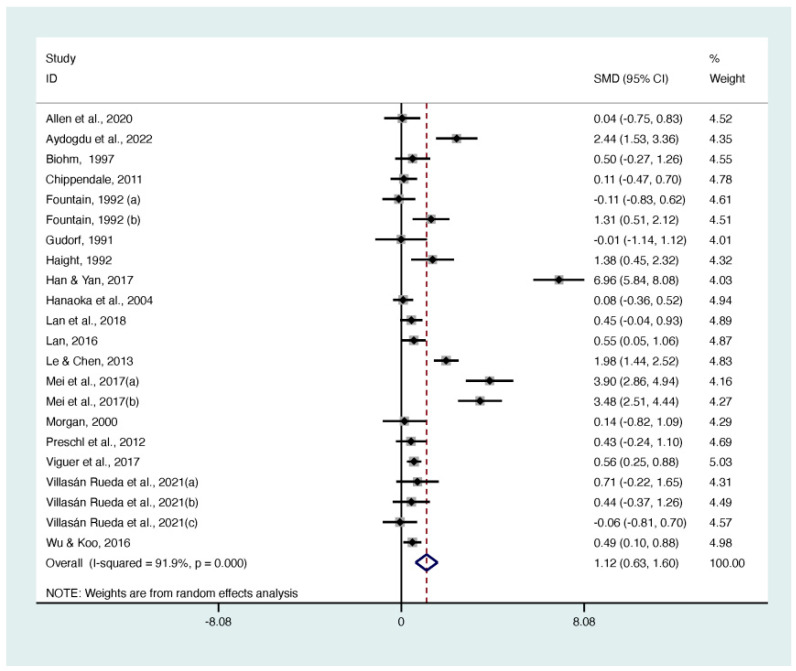
Forest plot of the effect on life satisfaction [20,21,22,24,26,28,29,30,31,33,34,35,40,41,43,48,49,51].

**Figure 5 behavsci-13-00830-f005:**
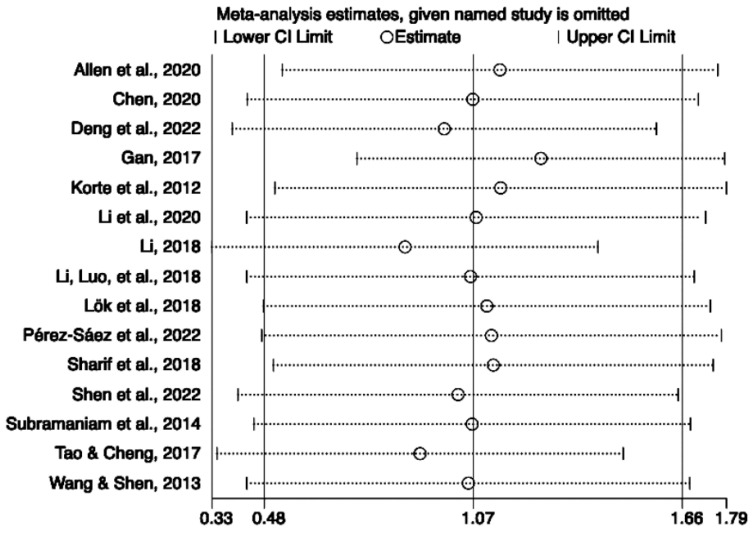
Sensitivity analysis of the included studies on quality of life [20,23,25,27,32,36,37,38,39,42,44,45,46,47,50].

**Figure 6 behavsci-13-00830-f006:**
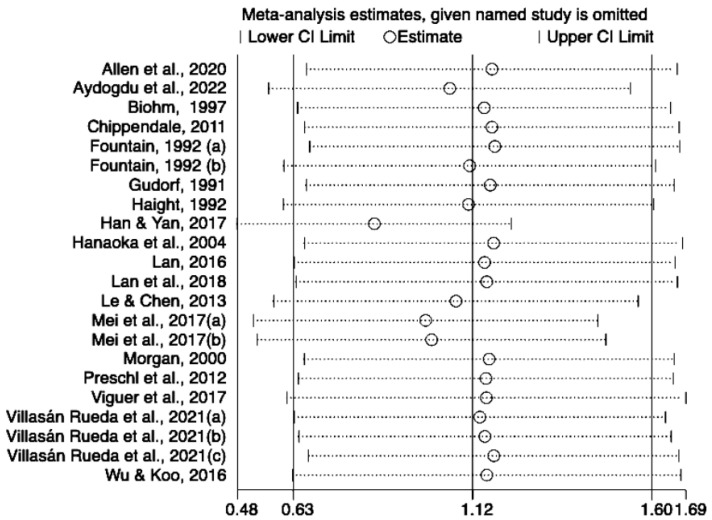
Sensitivity analysis of the included studies on life satisfaction [20,21,22,24,26,28,29,30,31,33,34,35,40,41,43,48,49,51].

**Figure 7 behavsci-13-00830-f007:**
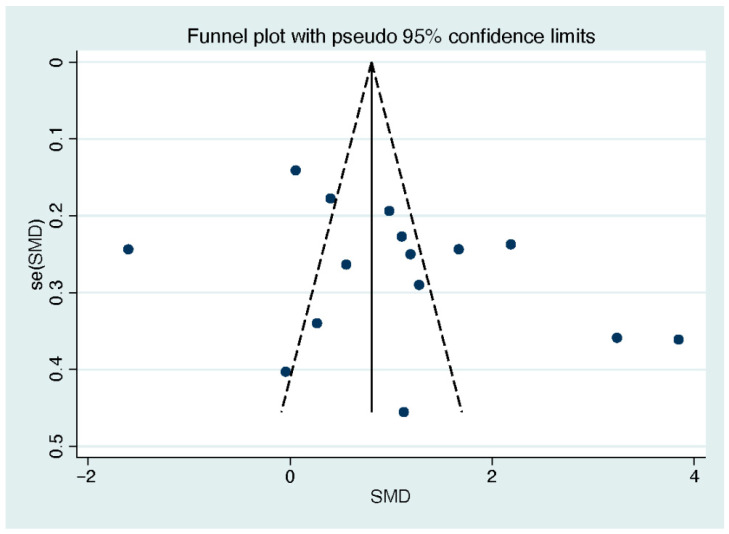
Funnel plot of the included studies on quality of life. Note. Each data point within the graph corresponds to an included study.

**Figure 8 behavsci-13-00830-f008:**
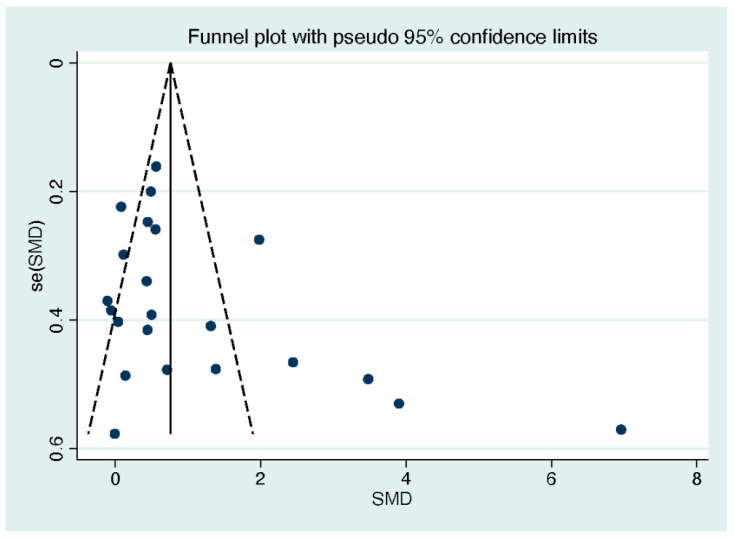
Funnel plot of the included studies on life satisfaction. Note. Each data point within the graph corresponds to an included study.

**Table 1 behavsci-13-00830-t001:** Study characteristics.

Study	Country	Condition	Participant				Intervention					Outcome (Scale)	Follow-Up
			N	Mean Age (SD)	Female (%)	Health Condition	GRP/IDV	Setting	Conductor	Duration	Content		
Allen et al., 2020 [20]	Ireland	RT	14	69.00 (4.84)	71.43	Healthy older adults	GRP	N/A	N/A	60 min/session; 6 weekly sessions	A semi-structured reminiscence program	QoL (CASP-19); LS (SWLS)	N/A
		Discussion	11	68.80 (4.97)	90.91	Healthy older adults	GRP	N/A	N/A	60 min/session; 6 weekly sessions	The present and future	QoL (CASP-19); LS (SWLS)	N/A
Aydogdu et al., 2022 [21]	Northern Cyprus	RT	17	76.17 (9.21)	64.71	Without hearing problems or cognitive impairment	GRP	Nursing homes	The researcher	45~60 min/session; 8 weekly sessions	Instrumental RT based on Roy’s adaptation model and group activities	LS (LSI-A)	N/A
		No treatment	17	73.94 (7.36)	41.18	Without hearing problems or cognitive impairment	N/A	N/A	N/A	N/A	N/A	LS (LSI-A)	N/A
Biohm, 1997 [22]	The United States	RT	14	N/A	93.00	Healthy older adults	GRP	Nursing homes	The investigator	45 min/session; twice a week; 4 weeks	Depicting major events from past decades by large posters	LS (LSI-A)	6 weeks
		Discussion	13	N/A	85.00	Healthy older adults	GRP	Nursing homes	The investigator	45 min/session; twice a week; 4 weeks	Depicting current events by large posters	LS (LSI-A)	6 weeks
Chen, 2020 [23]	China	RT	45	71.00 (5.30)	42.22	With stroke	IDV	Hospital	Nurse practitioner	60 min/session; 6 weekly sessions	Discussion around the theme of reminiscence	QoL (WHOQOL)	N/A
		Usual care	45	70.90 (5.20)	40.00	With stroke	N/A	N/A	N/A	N/A	Health care and education	QoL (WHOQOL)	N/A
Chippendale, 2011 [24]	The United States	LR	23	87.20 (6.30)	69.60	Without advanced disease	GRP	Senior residences	The researcher	90 min/session, 8 weekly sessions	Writing and reading life stories	LS (LSI-A)	N/A
		Waiting list	22	80.80 (7.50)	68.20	Without advanced disease	N/A	N/A	N/A	N/A	N/A	LS (LSI-A)	N/A
Deng et al., 2022 [25]	China	RT	58	70.31 (7.98)	44.83	With MI and depressive symptom after PCI	IDV	Hospital	The researcher	30~45 min/session; 3 sessions/week, 6 weeks	Discussion around the theme of reminiscence	QoL (GQOLI-74)	N/A
		Usual care	57	69.61 (7.32)	33.33	With MI and depressive symptom after PCI	IDV	Hospital	N/A	N/A	Cognitive intervention, medication, psychological support, and cardiac rehabilitation exercises	QoL (GQOLI-74)	N/A
Fountain, 1992 [26]	The United States	LR	27	N/A	89.00	With multiple diagnoses including mild Alzheimer’s disease, chronic brain syndrome, assorted cardiovascular conditions	GRP	Health care center and nursing home	The researcher and the second-year group work students	50~60 min/session; 12 bi-weekly sessions	Discussion around the theme of reminiscence	LS (RPGCMS)	N/A
		LR coupled with religious themes	25	N/A	84.00	With multiple diagnoses including mild Alzheimer’s disease, chronic brain syndrome, assorted cardiovascular conditions	GRP	Health care center and nursing home	The researcher and a local retired Lutheran pastor	12 bi-weekly sessions	Discussion around the theme of reminiscence by utilizing religion as principle	LS (RPGCMS)	N/A
		No treatment	20	N/A	85.00	With multiple diagnoses including mild Alzheimer’s disease, chronic brain syndrome, assorted cardiovascular conditions	N/A	N/A	N/A	N/A	N/A	LS (RPGCMS)	N/A
Gan, 2017 [27]	China	RT	45	71.50 (1.60)	57.78	With Alzheimer’s	IDV	Hospital	Professional nurses	N/A	Discussion around the theme of reminiscence	QoL (QOL-AD)	N/A
		Usual care	45	71.50 (1.30)	31.11	With Alzheimer’s	N/A	Hospital	Professional Nurses	N/A	Daily care	QoL (QOL-AD)	N/A
Gudorf, 1991 [28]	The United States	LR	6	83.5	100.00	With the ability to concentrate for 1 h	IDV	Home community	The researcher and the nursing home social worker	60 min/session; 8 bi-weekly sessions	Discussion around the theme of life stories	LS (LSES)	5 weeks
		No treatment	6	87.2	83.33	With the ability to concentrate for 1 h	N/A	N/A	N/A	N/A	N/A	LS (LSES)	5 weeks
Haight, 1992 [29]	The United States	LR	10	N/A	N/A	N/A	IDV	Home	The investigator and research assistants	60 min/session; 6 weekly sessions	Guide the older person through his or her memories by attentive listening	LS (LSI-A)	1 year
		A friendly visit	13	N/A	N/A	N/A	N/A	Home	N/A	N/A	N/A	LS (LSI-A)	1 year
Han & Yan, 2017 [30]	China	RT	45	N/A	N/A	With stroke	GRP	Hospital	Professional nurses	6 weekly sessions	Reminiscence activities with songs, films, photos, and souvenirs	LS (LSI-A)	N/A
		Usual care	45	N/A	N/A	With stroke	N/A	Hospital	N/A	N/A	Daily care and health education	LS (LSI-A)	N/A
Hanaoka et al., 2004 [31]	Japan	LR	42	81.62 (5.09)	85.71	No history of mental disorders; without cognitive impairment	GRP	N/A	Therapists	60 min/session; 8 weekly sessions	Discussion around the theme of reminiscence	LS (LSI-A)	3 months
		Discussion	38	81.97 (6.33)	86.84	No history of mental disorders; without cognitive impairment	N/A	N/A	Therapists	60 min/session; 8 weekly sessions	Discussion activities about health	LS (LSI-A)	3 months
Korte et al., 2012 [32]	Netherlands	LR	100	63.30 (6.20)	80.00	With slight to moderate depressive and anxiety symptoms	GRP	Mental health care institutions	Therapists	120 min/session, 8 sessions	Discussion around the theme of “the stories we live by”	QoL (EQ-5D)	3 months; 9 months
		Usual care	102	63.30 (6.80)	73.50	With slight to moderate depressive and anxiety symptoms	N/A	N/A	N/A	N/A	N/A	QoL (EQ-5D)	3 months
Lan, 2016 [33]	China	LR	31	83.06 (6.88)	64.52	With frailty	IDV	The nursing home	The researcher	30~60 min/session; 6 weekly sessions	Discussion around the theme of life stories through photographs	LS (LSI-A)	N/A
		Usual care	31	82.90 (7.10)	61.29	With frailty	N/A	N/A	N/A	N/A	N/A	LS (LSI-A)	N/A
Lan et al., 2018 [34]	China	LR	33	83.10 (6.50)	67.60	With frailty	IDV	The nursing home	The researcher	30~60 min/session; 6 weekly sessions	Discussion around the theme of life stories through photographs	LS (LSI-A)	N/A
		Usual care	34	83.50 (6.60)	62.20	With frailty	N/A	N/A	N/A	N/A	N/A	LS (LSI-A)	N/A
Le & Chen, 2013 [35]	China	RT	40	70.80 (16.20)	50.00	With mild or moderate depression	GRP	N/A	Professional nurses	40~50 min/session; 6 weekly sessions	Reminiscence activities with songs, films, photos, and souvenirs	LS (LSI-A)	N/A
		Usual care	40	72.00 (17.30)	55.00	With mild or moderate depression	N/A	N/A	N/A	N/A	Daily care and health education	LS (LSI-A)	N/A
Li, 2018 [36]	China	RT	45	63.70 (3.10)	46.67	With stroke	IDV	Hospital	A therapist	4 weekly sessions	Reminiscence activities with songs letters, and meaningful items	QoL (KPS)	N/A
		Usual care	45	63.30 (3.10)	44.44	With stroke	N/A	Hospital	Doctors and nurses	N/A	Medication and health education	QoL (KPS)	N/A
Li, Luo et al., 2018 [37]	China	RT	38	79.88 (5.56)	42.11	With Alzheimer’s	IDV	Hospital	The therapist	45 min/session; 10 weekly sessions	Discussion around the theme of reminiscence	QoL (QOL-AD)	N/A
		Usual care	38	80.22 (5.25)	52.63	With Alzheimer’s	N/A	Hospital	N/A	10 weeks	Daily treatment	QoL (QOL-AD)	N/A
Li et al., 2020 [38]	China	RT	60	69.89 (2.38)	40.00	With chronic diseases	IDV	Hospital	Nurses	N/A	Reminiscence activities with music, photos, and books	QoL (SF-36)	N/A
		Usual care	60	70.48 (2.35)	36.67	With chronic diseases	N/A	Hospital	Nurses	N/A	Daily care and support	QoL (SF-36)	N/A
Lök et al., 2019 [39]	Turkey	RT	30	N/A	60.00	With Alzheimer’s	GRP	The nursing home	N/A	60 min/session; 8 weekly sessions	Reminiscence activities with photographs, household goods, music, and foods	QoL (QOL-AD)	N/A
		Control	30	N/A	53.40	With Alzheimer’s	N/A	N/A	N/A	N/A	N/A	QoL (QOL-AD)	N/A
Mei et al., 2018 [40]	China	RT	44	69.67 (2.35)	80.00	With stroke	IDV	Caregivers’ homes	A psychologist	45~60 min/session; 8 weekly sessions	Reminiscence activities with diaries, letters, old photos, songs, and newspapers	LS (SWLS)	1 month; 3 months
		Health education	44	69.80 (5.58)	75.00	N/A	N/A	N/A	Community health workers	N/A	N/A	LS (SWLS)	1 month; 3 months
Morgan, 2000 [41]	The United Kingdom	LR	8	80.50 (5.75)	75.00	With mild to moderate dementia	IDV	Care homes	The therapist	60 min/session; 12 weekly sessions	Life story book	LS (LSI-A)	6 weeks
		No treatment	9	84.44 (7.81)	77.80	With mild to moderate dementia	N/A	N/A	N/A	N/A	N/A	LS (LSI-A)	6 weeks
Pérez-Sáez et al., 2022 [42]	Portugal	RT	62	82.39 (7.58)	71.60	With neurocognitive disorder	IDV	Social care institutions	Therapists	50 min/session; 26 bi-weekly sessions	Reminiscence activities with images, music, riddles, and theme worksheets	QoL (QOL-AD)	N/A
		Usual care	68	82.68 (7.32)	68.90	With neurocognitive disorder	N/A	N/A	N/A	N/A	N/A	QoL (QOL-AD)	N/A
Preschl et al., 2012 [43]	Switzerland	LR	20	72.50 (4.50)	75.00	With minimal or moderate depression	IDV	N/A	Therapists	60~90 min/session; 6 weekly sessions	Structured life review by face to face and computer form	LS (LSI-A)	3 months
		Waiting list	16	67.00 (3.10)	56.30	With minimal or moderate depression	N/A	N/A	N/A	six weeks in total	N/A	LS (LSI-A)	N/A
Sharif et al., 2018 [44]	Iran	LR	17	N/A	52.90	N/A	IDV	Day care centers	The researcher	120 min/session; 8 bi-weekly sessions	N/A	QoL (WHOQOL-BREF)	1 month
		Usual care	18	N/A	50.00	N/A	N/A	Day care centers	N/A	N/A	N/A	QoL (WHOQOL-BREF)	1 month
Shen et al., 2022 [45]	China	RT	46	73.74 (4.77)	43.48	With Alzheimer’s	IDV	Hospital	Therapists	40~60 min/session; 8 weekly sessions	Reminiscence activities with photos and meaningful items	QoL (QOL-AD)	N/A
		Usual care	46	73.70 (4.78)	41.30	With Alzheimer’s	N/A	Hospital	N/A	8 weeks	Daily care and health education	QoL (QOL-AD)	N/A
Subramaniam et al., 2014 [46]	The United Kingdom	LR	11	84.50 (6.70)	72.70	With dementia	IDV	Care homes	The therapist	60 min/session; 12 weekly sessions	developing life story book for themselves	QoL (QOL-AD)	6 weeks
		Storybook as gift	12	88.30 (6.00)	66.70	With dementia	N/A	Care homes	The therapist	5 or 6 times over 12 weeks	Researcher worked with the participant’s relative to develop a life story book	QoL (QOL-AD)	6 weeks
Tao & Cheng, 2017 [47]	China	RT	37	71.20 (5.30)	43.24	With dementia	IDV	Hospital	N/A	12 weekly sessions	Reminiscence activities with photos and meaningful items	QoL (QOL-AD)	N/A
		Usual care	37	70.80 (5.10)	45.95	With dementia	N/A	Hospital	N/A	N/A	Daily treatment	QoL (QOL-AD)	N/A
Viguer et al., 2017 [48]	The Dominican Republic	RT	80	73.06 (6.90)	56.30	Without cognitive impairment or clinical depression	GRP	N/A	A trained psychologist	120 min/session; 10 weekly sessions	Discussion around the theme of reminiscence	LS (LSI-A)	3 months
		Waiting list	80	71.44 (6.30)	51.20	Without cognitive impairment or clinical depression	N/A	N/A	N/A	N/A	N/A	LS (LSI-A)	3 months
Villasán Rueda et al., 2021 [49]	Spain	RT	14	N/A	N/A	Healthy aged	GRP	Day care centers	N/A	60 min/session; 12 bi-weekly sessions	Positive reminiscence	LS (LSI-A)	N/A
		Cognitive stimulation	13	N/A	N/A	Healthy aged	GRP	Social care institutions	N/A	60 min/session; 12 bi-weekly sessions	Attention, perception, memory, language, inhibition, planning, reasoning, calculation, and drawing trainings	LS (LSI-A)	N/A
		RT	11	N/A	N/A	With mild cognitive impairment	GRP	Social care institutions	N/A	60 min/session; 12 bi-weekly sessions	Positive reminiscence	LS (LSI-A)	N/A
		Cognitive stimulation	13	N/A	N/A	With mild cognitive impairment	GRP	Social care institutions	N/A	60 min/session; 12 bi-weekly sessions	Attention, perception, memory, language, inhibition, planning, reasoning, calculation, and drawing trainings	LS (LSI-A)	N/A
		RT	20	N/A	N/A	With Alzheimer’s	GRP	Social care institutions	N/A	60 min/session; 12 bi-weekly sessions	Positive reminiscence	LS (LSI-A)	N/A
		Cognitive stimulation	6	N/A	N/A	With Alzheimer’s	GRP	Social care institutions	N/A	60 min/session; 12 bi-weekly sessions	Attention, perception, memory, language, inhibition, planning, reasoning, calculation, and drawing trainings	LS (LSI-A)	N/A
Wang & Shen, 2013 [50]	China	LR	29	83.00 (6.91)	N/A	With Alzheimer’s, vascular dementia or mixed dementia	IDV	Hospital	Nurses	30~45 min/session; 8 weekly sessions	Life story book	QoL (DQOL)	4 weeks
		Usual care	29	82.40 (6.84)	N/A	With Alzheimer’s, vascular dementia or mixed dementia	N/A	Hospital	Nurses	N/A	Daily treatment	QoL (DQOL)	4 weeks
Wu & Koo, 2016 [51]	China	RT	53	73.50 (7.30)	64.20	With mild or moderate dementia	GRP	Hospital	The researcher	60 min/session; 6 weekly sessions	Spiritual reminiscence activities	LS (LSI-A)	N/A
		Interview	50	73.60 (7.60)	74.00	With mild or moderate dementia	N/A	N/A	The researcher	N/A	N/A	LS (LSI-A)	N/A

Note. CASP-19 = Control, Autonomy, Self-realisation and Pleasure Quality of Life questionnaire; DQOL = Dementia Quality of Life Instrument; EQ-5D = Euroqol questionnaire; GQOLI-74 = Generic Quality of Life Inventory-74; GRP = group intervention; IDV = individual intervention; KPS = Karnofsky; KHRS = King & Hunt Religiosity Scale; LIS-A = Life Satisfaction Index A; LR = life review; LS = life satisfaction; LSES = Life Satisfaction in the Elderly Scale; N/A = no related information is available; N = sample size; QoL = quality of life; QOL-AD = Quality of Life in Alzheimer’s Disease scale; RPGCMS = Revised Philadelphia Geriatric Center Morale Scale; RT = reminiscence therapy; SF-36 = the MOS item short from health survey; SWLS = Satisfaction With Life Scale; W% = the percentage of female participants; WHOQOL = World Health Organization Quality of Life; WHOQOL-BREF = World Health Organization Quality of Life Questionnaire.

**Table 2 behavsci-13-00830-t002:** Subgroup analyses of quality of life.

	Studies (n)	Participants (n)	I^2^	SMD [95% CI]	*p* (Overall Effects)	*p* (Group Differences)
**Form of intervention**						0.006
Group	3	287	34.40%	0.18 [−0.15, 0.52]	0.28	
Individual	12	993	95.80%	1.29 [0.58, 2.01]	<0.001	
**Total sessions of intervention**						0.71
≤8 sessions	9	767	94.80%	1.21 [0.48, 1.94]	<0.001	
>8 sessions	4	303	94.10%	1.47 [0.33, 2.61]	<0.001	

I^2^: heterogeneity; SMD: standardized mean difference.

**Table 3 behavsci-13-00830-t003:** Subgroup analyses of life satisfaction.

	Studies (n)	Participants (n)	I^2^	SMD [95% CI]	*p* (Overall Effects)	*p* (Group Differences)
**Form of intervention**						0.69
Group	14	793	92.90%	1.04 [0.43, 1.65]	<0.001	
Individual	8	305	90.30%	1.26 [0.40, 2.12]	<0.001	
**Total sessions of intervention**						0.006
≤8 sessions	15	772	94.20%	1.46 [0.76, 2.15]	<0.001	
>8 sessions	7	326	37.50%	0.44 [0.10, 0.77]	0.14	
**Comparison**						0.06
Passive control	8	376	74.20%	0.59 [0.12, 1.07]	<0.001	
Active control	14	722	94.20%	1.43 [0.70, 2.16]	<0.001	

Note. I^2^, heterogeneity; SMD, standardized mean difference.

## Data Availability

Not applicable.

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
