# Peer review of "Effectiveness on Quality of Life and Life Satisfaction for Older Adults: A Systematic Review and Meta-Analysis of Life Review and Reminiscence Therapy across Settings"

_behavsci, 2023, doi:10.3390/bs13100830_

Round 1
Reviewer 1 Report
Title
The title effectively conveys the topic of the research, which is important for readers to understand the subject matter. It mentions the key elements: "Quality of Life," "Life Satisfaction," "Older Adults," and "Systematic Review and Meta-Analysis."
Abstract Review:
The abstract provides a concise overview of the research, including its background, objectives, methods, results, and conclusions. It effectively summarizes the key findings of the study. However, there are a few areas for improvement:
· Inclusion of Key Variables: Mention the key variables or factors that were assessed during the interventions. This would provide readers with a better understanding of the study's scope.
Introduction
The introduction provides a comprehensive background on the aging population and the concept of active aging, leading to the focus on life review and reminiscence therapy as interventions for older adults. Overall, it sets the stage for the research well, but there are some areas for improvement:
1. Clarity and Conciseness:
The introduction begins with important statistics about the aging population, but it could benefit from a more concise presentation of these facts. Try to summarize the key statistics in a more straightforward manner.
Consider organizing the introductory sentences to flow more smoothly and logically, making it easier for readers to follow the narrative.
2. Objective Clarity:
While the introduction mentions the objective, it could be made more explicit. Clearly state the main research question or hypothesis that the study aims to address. This will help readers understand the specific focus of the research.
3. Rationale and Gap Identification:
The introduction mentions the conflicting findings regarding the effectiveness of life review and reminiscence therapy, which is crucial. However, it would be helpful to explicitly state that these discrepancies create a research gap that this study intends to address.
4. Structure and Flow:
Consider restructuring the introduction to create a more logical flow. Start with a concise presentation of the aging population statistics, move on to the concept of active aging, and then introduce the role of psychological interventions, specifically life review and reminiscence therapy. This can provide a smoother transition to the research objective.
Methods
The methods section provides a clear overview of the systematic review and meta-analysis process. Overall, it is well-structured and detailed. Here are some observations and suggestions for improvement:
1. Clarity and Transparency:
The section effectively communicates the steps taken in conducting the systematic review and meta-analysis, aligning with PRISMA guidelines. It also highlights the registration of the review protocol with PROSPERO, demonstrating transparency.
2. Search Strategy:
The inclusion of eight databases for the systematic literature search enhances the comprehensiveness of the review. However, it would be helpful to briefly mention the specific time frame of the search, such as the years considered for inclusion.
3. Inclusion and Exclusion Criteria:
The inclusion and exclusion criteria are clearly defined, making it easy to understand which studies were considered for inclusion. However, it would be beneficial to mention the rationale behind certain criteria, such as why studies with participants aged 55 and above were selected.
4. Data Extraction:
The process of data extraction is well-described. It includes details on how data were coded and handled when discrepancies arose. However, it would be helpful to mention any specific variables that were extracted, such as primary outcomes or demographic data.
5. Quality Assessment:
The use of the Cochrane risk-of-bias tool (RoB2) for assessing the risk of bias in included articles is appropriate. However, it might be beneficial to briefly explain the key domains assessed by RoB2 (randomization process, intervention deviation, etc.) for readers who may not be familiar with the tool.
Results
The results section of the article presents the findings of the systematic review and meta-analysis. Overall, the section provides a clear presentation of the study selection process, characteristics of the included studies, main outcomes, sensitivity analysis, and subgroup analyses. Here are some observations and suggestions for improvement:
1. Study Selection:
The presentation of the study selection process is clear and follows the PRISMA guidelines. It effectively conveys how the final set of studies for the meta-analysis was chosen.
Study Characteristics:
The section provides detailed information about the included studies, such as sample sizes, participant demographics, intervention details, and risk of bias assessment. This comprehensive information is valuable for readers.
3. Main Outcomes:
The presentation of main outcomes related to quality of life and life satisfaction is clear and includes relevant statistical information (effect sizes, confidence intervals, heterogeneity, and p-values).
4. Sensitivity Analysis:
The description of sensitivity analysis is appropriate and serves the purpose of identifying potential outliers and assessing the robustness of the findings.
The section provides clear information on which studies were removed during sensitivity analysis and the impact on the results.
5. Meta-Regression and Subgroup Analysis:
The use of meta-regression and subgroup analysis to explore sources of heterogeneity is appropriate and enhances the understanding of the factors influencing the outcomes.
The results of the meta-regression and subgroup analyses are clearly presented, including statistical significance.
6. Publication Bias:
The assessment of publication bias using funnel plots and Egger's regression intercept is a valuable addition to the analysis, as it addresses potential bias in the included studies.
9. Clarity and Consistency:
Ensure consistent formatting of results, including effect size format (e.g., SMD 1.07) and confidence intervals (e.g., 95% CI 0.48 to 1.66).
Discussion
The discussion section of the article provides an insightful analysis of the main findings of the meta-analysis on life review and reminiscence therapy for older adults. It also discusses implications for future studies. Here are some observations and suggestions for improvement:
1. Main Findings:
The discussion effectively highlights the significant improvements in quality of life and life satisfaction observed in older adults following life review or reminiscence therapy. This clear summary of the main findings aligns with the study's objectives.
2. Long-Term Effects:
The mention of the limited evaluation of the long-term impact of life review and reminiscence therapy on quality of life is valuable. However, it would be beneficial to briefly discuss the potential reasons for this limitation, such as the lack of long-term follow-up data in the included studies or the challenges of assessing sustained effects over time.
3. Heterogeneity and Outliers:
The acknowledgment of a high level of heterogeneity and the identification of outliers in certain studies are important. Discussing potential reasons for this heterogeneity, such as differences in intervention protocols, participant characteristics, or measurement tools, would enhance the discussion's depth.
4. Variability in Outcome Concept:
The discussion mentions the multifaceted nature of quality of life and the different measurement tools used. To strengthen this point, you could provide specific examples of how variations in measurement tools might have influenced the results or interpretations.
5. Implications for Future Studies:
The discussion regarding subgroup analyses related to intervention settings and duration is informative. However, it would be beneficial to discuss the practical implications of these findings for researchers and healthcare providers. For example, how can these insights guide the design of future interventions or clinical practices?
6. Ethical Considerations:
The mention of potential risks of bias related to informed consent and experimental intent is important. Expanding on this point by discussing ethical considerations when working with older adults in research, particularly those with cognitive impairments, would add depth to the discussion.
7. Mechanisms and Study Design:
The discussion alludes to the need for more rigorous study designs and the importance of reporting comprehensive study design details. Expanding on this by suggesting specific improvements in study design, such as blinding or randomization procedures, would be valuable.
8. Strengths and Limitations:
The discussion section effectively highlights both the strengths and limitations of the study. To enhance this section, you could briefly discuss how the study's focus on improving positive emotions aligns with the broader goals of active aging and overall well-being.
9. Language Limitation:
When discussing the limitation of including only English and Chinese studies, you may consider mentioning that language restrictions are common in systematic reviews and meta-analyses due to practical constraints.
10. Future Research Directions:
- While you touch upon the need for more high-quality studies in the future, you could provide specific recommendations for future research directions in the field of life review and reminiscence therapy, such as exploring potential moderators or mediators of treatment effects.
Overall, the discussion section effectively addresses the study's main findings, limitations, and implications for future research. Expanding on some of the points mentioned and providing practical recommendations for researchers and practitioners would further enhance the discussion.
Reviewer 2 Report
see the attached file

see comments in the attached file.
Reviewer 3 Report
The manuscript contains the results of a meta analysis and systematic review of reminiscence therapy and life review of studies published in English and Chinese. The were 32 studies included for a total of 2353 participants. It was found that therapy led to an improvement in life satisfaction and quality of life. The manuscript is well written.
Comments:
It is surprising that interventions with more than 6-8 sessions would lead to no significant effect. It would be expected that perhaps the effect would reach a plateau – why would there seemingly be a decrease in the effectiveness of the intervention?
